# Advanced Biotechnological Interventions in Mitigating Drought Stress in Plants

**DOI:** 10.3390/plants13050717

**Published:** 2024-03-04

**Authors:** Özhan Şimşek, Musab A. Isak, Dicle Dönmez, Akife Dalda Şekerci, Tolga İzgü, Yıldız Aka Kaçar

**Affiliations:** 1Horticulture Department, Agriculture Faculty, Erciyes University, Kayseri 38030, Türkiye; akidal_@hotmail.com; 2Agricultural Sciences and Technology Department, Graduate School of Natural and Applied Sciences, Erciyes University, Kayseri 38030, Türkiye; musabisak11@gmail.com; 3Biotechnology Research and Application Center, Çukurova University, Adana 01330, Türkiye; dicledonmez4@gmail.com; 4National Research Council of Italy (CNR), Institute of BioEconomy, 50019 Florence, Italy; tolga.izgu@ibe.cnr.it; 5Horticulture Department, Agriculture Faculty, Çukurova University, Adana 01330, Türkiye; ykacar@cu.edu.tr

**Keywords:** biotechnological interventions, CRISPR-Cas9, genome editing, drought stress, microbial biotechnology, plant genomics

## Abstract

This comprehensive article critically analyzes the advanced biotechnological strategies to mitigate plant drought stress. It encompasses an in-depth exploration of the latest developments in plant genomics, proteomics, and metabolomics, shedding light on the complex molecular mechanisms that plants employ to combat drought stress. The study also emphasizes the significant advancements in genetic engineering techniques, particularly CRISPR-Cas9 genome editing, which have revolutionized the creation of drought-resistant crop varieties. Furthermore, the article explores microbial biotechnology’s pivotal role, such as plant growth-promoting rhizobacteria (PGPR) and mycorrhizae, in enhancing plant resilience against drought conditions. The integration of these cutting-edge biotechnological interventions with traditional breeding methods is presented as a holistic approach for fortifying crops against drought stress. This integration addresses immediate agricultural needs and contributes significantly to sustainable agriculture, ensuring food security in the face of escalating climate change challenges.

## 1. Introduction

Drought represents a significant global challenge, impacting various societal aspects, including agriculture, water security, and the economy. It is a phenomenon that transcends geographical boundaries, affecting arid, semi-arid, and even temperate regions. The consequences of drought are multifaceted, ranging from diminished agricultural production to economic strain and food security issues, as evidenced by studies such as Haile et al. [1], Leng et al. [2], and El-Hashash et al. [3]. Additionally, Shahbazbegian and Bagheri [4] and Krupa et al. [5] highlight the socio-economic repercussions of these agricultural impacts. The escalation in the severity of drought conditions can be attributed to climate change and anthropogenic warming, leading to an increase in the frequency, magnitude, and intensity of drought events globally [6,7,8]. This exacerbation of drought conditions extends beyond environmental and economic impacts, affecting human health and mental well-being, and increasing social vulnerability, as noted by Vins et al. [9] and Ascott et al. [10]. The complexity in assessing drought risk arises from the interplay between drought probability, socio-economic consequences, and the exposure and vulnerability of water users to water shortage, a concept elaborated by Mens et al. [11].

Moreover, the issue of drought has cross-sectoral implications, influencing various societal aspects such as agriculture, water resources, and even human migration, as discussed by Conradt et al. [12]. The far-reaching consequences of drought are not limited to economic sectors but also initiate dynamic mechanisms with socio-environmental consequences, impacting the sustainability of affected areas [4]. In summary, the global challenge posed by drought is extensive and multifaceted, necessitating a comprehensive approach to drought risk assessment and adaptive policymaking for effectively addressing its complex consequences on a worldwide scale. A crucial aspect in combating the effects of drought is enhancing drought tolerance in plants, particularly in the context of sustainable agriculture amid increasing climate variability and water scarcity. Plant resilience to drought stress is vital for maintaining agricultural productivity and food security [13]. This enhancement involves a multifaceted approach, including developing more drought-tolerant cultivars and the sustainable adoption of agronomic practices like water-saving irrigation. Additionally, the utilization of plant growth-promoting rhizobacteria to alleviate drought stress is also significant, as indicated in studies by Zipper et al. [14] and Niu et al. [15].

## 2. Understanding Plant Responses to Drought Stress

On a molecular level, the role of transcription factors and network candidate genes is pivotal in augmenting the drought tolerance potential in crop plants [16,17]. Understanding plant adaptive responses to drought is imperative for improving breeding strategies for drought-tolerant crops [18]. Drought is the most significant environmental stress in agriculture globally, and enhancing yield in water-limited environments is a primary goal of plant breeding [19]. Strategies such as developing and deploying drought-tolerant maize varieties are crucial in mitigating the detrimental effects of drought on agricultural productivity and livelihoods [20].

Furthermore, selecting drought-tolerant crop types and managing single and double crops during dry periods is essential for water conservation and sustainable water management in agriculture [21]. The role of micronutrients in enhancing drought tolerance and supporting sustainable plant growth under drought conditions has been highlighted, with nutrients found in plants like *Moringa oleifera* playing a crucial role in osmoregulation and improving crop drought tolerance [22]. The application of biostimulants is also an innovative approach for achieving food security under drought conditions, underscoring the immediate challenge for the scientific community to develop resilience against abiotic stresses, specifically drought [23].

Significant research has focused on identifying specific genes and their functions in response to water-deficit conditions to understand and improve plant drought tolerance. Table 1 compiles recent findings from various studies that have explored the genetic responses of different plant species to drought stress. This table includes a range of genes from different families and their observed effects on plant species such as *Dactylis glomerata*, *Glycine max*, and *Arabidopsis thaliana*, among others. The table provides a comprehensive overview of how these genes contribute to drought tolerance through post-transcriptional gene regulation, tolerance to abiotic stresses, and scavenging reactive oxygen species. These insights are pivotal for developing strategies to enhance crop drought resilience, as they offer a deeper understanding of the molecular basis of drought response in plants.

In conclusion, enhancing plants’ drought tolerance for sustainable agriculture is paramount and requires a multidisciplinary approach, encompassing genetic, molecular, and agronomic strategies, to develop and deploy drought-tolerant crop varieties. Such efforts are critical for enhancing agricultural resilience due to climate variability and water scarcity. Improving drought resistance in crops is a crucial objective in agricultural biotechnology and involves integrating agronomic and biotechnological strategies, along with advanced genome editing tools, and employing plant growth regulators and rhizobacteria for the development of drought-tolerant cultivars [33,34].

Plants’ physiological and molecular responses to a water deficit constitute a complex and dynamic interplay of various mechanisms. This intricate response system has been the subject of extensive research, significantly contributing to crop management and breeding strategy advancements. Plants exhibit a range of adaptive responses at both cellular and whole-organism levels to cope with drought stress, making it a multifaceted phenomenon. These responses are broadly categorized into morphological, physiological, and biochemical changes. For instance, studies have documented the intricate molecular responses of plants under water-deficit conditions, including the activation of specific genes that protect cells against water scarcity and regulate the drought response [35,36].

Physiological adaptations, including modifications in stomatal conductance, photosynthetic efficiency, and the activation of antioxidant systems, play a crucial role in sustaining plant growth and productivity under conditions of water scarcity [37,38]. The investigation into differential gene expression in response to water-deficit stress is pivotal for devising molecular interventions aimed at bolstering water-stress resilience in plants [39]. Despite a wealth of studies on drought stress, there remains a gap in our understanding of the precise molecular mechanisms that plants employ to cope with a water deficit [40]. Identifying and elucidating these mechanisms, from osmotic regulation and stomatal behavior adjustments to root system modifications, are essential steps toward enhancing drought tolerance in crops. Krannich et al. [41] highlighted the ongoing research in cereals for understanding drought tolerance mechanisms. Oren et al. [42] discussed the significance of stomatal conductance in drought tolerance, linking it with the plant’s ability to regulate water loss. Additionally, research by Iseki et al. [43] and Chen et al. [44] emphasized the importance of root morphology and development in drought tolerance mechanisms. Bapela et al. [45] stressed the need to understand genetic, physiological, biochemical, and environmental interactions for improving drought tolerance.

Moreover, the molecular response of plants to drought stress involves a complex network of genes and proteins regulating various physiological and biochemical processes and includes the role of transcription factors, enzymes, and molecular chaperones in drought tolerance [46,47]. Studies have also shown the involvement of specific proteins in plants’ adaptation to stress conditions, such as the response of *A. thaliana* to a combination of drought and high temperatures [48]. The importance of post-translational modifications in drought response, as evidenced by protein phosphorylation associated with enhanced heat tolerance in response to drought stress, further highlights the complexity of these responses [49]. In summary, plants’ physiological and molecular responses to water deficit are intricate and involve various adaptive mechanisms at various levels. Understanding these responses is crucial for developing effective crop management and breeding strategies to enhance drought tolerance and ensure plant survival and productivity under water-limited conditions.

In response to drought conditions, plants undergo a range of physiological and molecular adaptations to mitigate the effects of water scarcity. These responses, summarized in Table 1, include mechanisms such as stomatal closure, alterations in root system and leaf morphology, osmotic adjustment, and changes in photosynthetic activity. Furthermore, plants exhibit a complex network of gene regulation, protein synthesis adjustments, enhanced ROS scavenging, and accumulation of secondary metabolites at the molecular level. These adaptive responses collectively enable plants to maintain cellular integrity and survive under limited water availability, as described in Table 2.

## 3. Traditional Breeding vs. Biotechnological Approaches

Traditional breeding, also known as cross-breeding, has enhanced agricultural productivity by combining desirable traits via sexual recombination. This method, dating back centuries, has played a critical role in developing numerous crop varieties adapted to various environmental conditions, including drought and salinity [50,51]. However, the traditional breeding approach has limitations, especially in modern agricultural demands and challenges. One of the primary limitations of conventional breeding is its reliance on crossing germplasm or random mutagenesis, a time-consuming and often inefficient process, especially when dealing with complex traits like drought tolerance [52,53]. Many plants’ slow growth and long life cycles further compound these challenges, making it difficult to quickly generate new varieties with the desired traits [54,55].

Furthermore, traditional mutagenesis can produce undesirable knockout mutations, reducing the effectiveness of this breeding approach [56]. Additionally, the selection of superior genotypes based on phenotypic traits has been limited due to low heritability, genetic interactions such as epistasis, and environmental–genotype interactions [57]. In response to these limitations, there has been a shift towards integrating genomics and genome editing techniques with traditional breeding methods. These advanced techniques offer a powerful tool for accelerating breeding and enhancing the efficiency of developing new crop varieties [58,59]. Genome editing, in particular, has emerged as a complementary method to traditional breeding, enabling the modification of specific traits with greater precision and in a shorter time frame. Drought tolerance breeding has significantly improved by integrating physiological and molecular breeding approaches. For instance, integrated genomics, physiology, and breeding strategies have been suggested to develop drought-tolerant cultivars, enhancing food security in a changing and more variable climate [60,61]. Physio-morphological trait-based approaches, such as the use of physiological traits in breeding for drought tolerance in wheat, have been highlighted for their potential in this area [62].

Additionally, exploring drought-tolerant, underutilized crops and using traditional landraces from drier regions offer new avenues for enhancing resilience to water deficiency [63,64]. Recent advances in genetic techniques, genomics tools, and breeding methodologies have opened promising avenues for identifying candidate genes and metabolic pathways underlying crop drought tolerance [65]. Marker-assisted breeding and the use of “omics” technologies have been recognized as effective for breeding drought tolerance, facilitating the identification of drought-related quantitative trait loci (QTLs) and enabling more efficient drought screening techniques [57,66]. In conclusion, while traditional breeding methods have historically played a pivotal role in crop improvement, their limitations in addressing complex traits like drought tolerance have led to the exploration and integration of modern biotechnological tools and genomics approaches. This synthesis of traditional and contemporary techniques presents promising prospects for improving crop drought tolerance, thereby contributing to sustainable agricultural practices and food security in the face of global climate challenges.

In addressing the challenges of crop improvement, both traditional breeding methods and modern biotechnological approaches offer unique advantages and limitations. As summarized in Table 2, conventional breeding methods rely on cross-breeding and selection based on phenotypic traits, which, while less resource-intensive, are often time-consuming and limited by genetic variability. In contrast, biotechnological approaches, such as gene editing and marker-assisted selection, offer a more precise and rapid development of crop varieties. These methods enable the introduction of novel traits, including enhanced stress resistance, that may not be available within the existing gene pool. However, they also raise potential ecological and ethical concerns. The comparison outlined in Table 3 highlights these differences, underscoring the need for a balanced and integrated approach to crop improvement in modern agriculture.

## 4. Genetic Engineering in Drought Stress Management

The development of genetically modified organisms (GMOs) for agricultural enhancement has embraced a variety of genetic engineering techniques, including transgenesis, cisgenesis, intragenesis, and genome editing [67,68]. Among these, the modification of regulatory genes, which encode proteins pivotal in plant stress responses, has emerged as a particularly effective strategy for crafting crops resilient to drought conditions. This approach leverages the intrinsic capacity of plants to adapt to environmental stressors by modulating key regulatory pathways. In pursuit of this goal, scientists have ventured into the realm of extremophiles, such as halophytes and thermophiles, organisms that thrive in harsh environments, to source heterologous genes that confer stress tolerance. The incorporation of these genes into the genomes of agricultural crops aims to endow them with enhanced survival mechanisms under adverse conditions. Notably, a spectrum of crop species, including rice, wheat, maize, mustard, soybean, sugarcane, tobacco, cotton, banana, and potato, have been the focus of genetic modifications to develop strains with improved drought tolerance. Illustrative of this innovative genetic engineering is the application of bacterial cold shock proteins (csp), which have been integrated into *Arabidopsis*, rice, and maize to bolster their drought stress resilience. Similarly, the transcription factor Hahb-4, derived from sunflower, has been successfully utilized to induce drought tolerance in soybeans. Additionally, genes from *E. coli* and *Rhizobium meliloti* have been employed to enhance stress tolerance in sugarcane. These examples underscore the potential of leveraging genetic diversity across the biological spectrum to fortify crops against the growing challenge of climate-induced stress, particularly drought, thereby ensuring agricultural productivity and food security in the face of climate change [68].

Recent advancements in understanding the molecular mechanisms underlying drought stress tolerance in transgenic plants have marked a significant milestone in agricultural biotechnology. Studies have elucidated how the overexpression of specific genes enhances drought resistance, offering new insights into the genetic manipulation of crops for improved resilience to environmental stresses. For example, the research of Todaka et al. [69] on the ZmGF14-6 gene in transgenic rice and Jiang et al. [70] on OsSNAC1 in transgenic rice and cotton plants underscore the role of specific genes in augmenting drought tolerance. These studies reveal that such genetic modifications can regulate crucial processes like ROS homeostasis, stomatal closure, and transpiration rate, thereby enhancing the plant’s ability to withstand water scarcity. Similarly, the research by Yang et al. [71] on OsbZIP23 in transgenic rice highlights the potential of genetic engineering in improving drought resistance and salt tolerance, showcasing the multifaceted benefits of this approach.

Additionally, the heterologous expression of genes from different species, such as the research of Li et al. [72] with MsGME from alfalfa in Arabidopsis, demonstrates the versatility and potential of transgenic approaches in conferring stress tolerance across various plant species. This cross-species gene expression can improve drought and salt stress resistance, thereby broadening the scope of genetic engineering in agriculture. Applying transgenic approaches goes beyond enhancing stress tolerance; it also encompasses maintaining or improving key agronomic traits. Moon et al. [73] and Ruan et al. [74] have shown that transgenic modifications can enhance drought tolerance in potatoes and sweet potatoes without compromising essential characteristics like yield. This balance is critical for the practical application of transgenic crops in agriculture.

Transgenic technology, while a powerful tool for crop improvement, is known for its high cost, time-intensive processes, and complexity. Moreover, its success rate varies across different cultivated crops, with many important ones showing limited responsiveness [75]. To address these challenges, researchers have explored alternative techniques such as acetylation [76], methylation [77], and ubiquitination [78], which play crucial roles in the development of drought-tolerant genotypes. Alongside these, newer, more efficient, and faster technologies for crop improvement and specific gene analysis have emerged, notably Virus-Induced Gene Silencing (VIGS) and CRISPR/Cas9 systems [79,80], offering promising avenues for functional genomics and targeted gene manipulation. VIGS, an RNA-based defense strategy originally against viruses, leverages virus vectors to target corresponding mRNAs, thus enabling the study of gene function in plants. This method has found extensive application in functional genomics through the engineering of various RNA viruses as vectors for abiotic stress gene analysis [81]. For instance, the knock-down of the drought-inducible variant H1-S in tomato has resulted in increased drought tolerance and enhanced stomatal closure [82]. Similarly, the overexpression of the TaH2A.7 variant in *Arabidopsis* has been linked to improved drought responses and reduced water loss [83]. Targeting the TaH2B-7D gene in wheat via VIGS has been shown to increase drought tolerance by affecting the relative electrolyte leakage rate and malondialdehyde levels, while also impacting proline and relative water content. These knock-down plants exhibited dwarf phenotypes and wilting symptoms compared to their non-modified counterparts, underscoring the role of the TaH2B-7D gene in drought resistance [84]. Additionally, the upregulation of the AtHUB2 gene in cotton has enhanced its drought response [77], whereas silencing genes like SpMAPK1, SpMAPK2, and SpMAPK3 in *Solanum pimpinellifolium*, GhWRKY27a in cotton, and others have diversely impacted drought tolerance, demonstrating the nuanced interplay of genetic factors in plant stress responses [85,86,87,88]. These advancements highlight the dynamic landscape of crop genetic improvement, showcasing the potential of both established and emerging genetic technologies in enhancing plant resilience against abiotic stresses.

Furthermore, the research community has explored the overexpression of transcription factors (TFs) and their role in drought stress management. The overexpression of specific TFs, such as DREB1A, as demonstrated by Karaba et al. [89], and ZmPTF1, as shown by Xue et al. [90], has been instrumental in activating stress response pathways, thereby enhancing drought tolerance. These TFs regulate various target genes, pivotal in the plant’s adaptive response to drought. The regulation of gene expression in response to drought involves a complex network of TFs that interact with various signaling pathways and undergo epigenetic regulation. The involvement of TFs in both ABA-dependent and ABA-independent regulatory systems, as highlighted in studies by Soma et al. [91] and Tang [92], underlines the multifaceted nature of TF-mediated drought responses.

In summary, recent research has provided valuable insights into the molecular basis of plant drought tolerance, emphasizing the potential of genetic engineering and the overexpression of specific genes and TFs. These advances contribute to our fundamental understanding of plant biology and offer practical solutions for developing drought-resistant crops, thereby contributing to sustainable agriculture and food security in the face of climate variability and water scarcity. The continued exploration of these molecular pathways and the integration of genomics with traditional breeding methods are promising for future crop improvement and stress tolerance developments.

## 5. CRISPR-Cas9 and Genome Editing

The CRISPR-Cas9 system, a groundbreaking technology initially discovered as a part of the adaptive immune system in archaea and bacteria, has revolutionized the field of plant biotechnology. Its ability to induce double-strand breaks (DSBs) in DNA at specific loci has made it an invaluable tool for precise genome editing in various crop species. Since its first application in plants in 2013, CRISPR-Cas9 has facilitated targeted gene editing and has been instrumental in introducing valuable agricultural traits into a wide range of crops, marking a significant advancement in agricultural biotechnology. One of the most notable applications of CRISPR-Cas9 in agriculture is the development of drought-resistant plant varieties. This technology has shown immense potential in enhancing plant resilience against abiotic stresses such as drought and salinity. CRISPR/Cas9 technology has emerged as a pivotal tool in the realm of plant biotechnology, offering unprecedented precision in targeted mutagenesis, which is instrumental in enhancing plant tolerance to various stressors, including drought. For instance, the knockout of specific genes using CRISPR-Cas9 has improved plant drought and salinity tolerance, as Chen et al. [93] demonstrated. This indicates that CRISPR-Cas9 can strategically modify plant genomes to cope better with environmental stresses, thereby contributing to sustainable agriculture. Beyond abiotic stress tolerance, CRISPR-Cas9 has also effectively enhanced resistance to various plant diseases, including destructive plant viruses. The system’s capability to target multiple DNA viruses simultaneously, as shown by Ali et al. [94], extends its utility in producing plants resistant to a range of viral infections. This feature is particularly beneficial for safeguarding crops against multiple pathogenic threats. However, it is important to note that the outcomes of CRISPR-Cas9 editing can vary. For example, Wang et al. [54] observed reduced drought tolerance in CRISPR/Cas9-mediated slmapk3 mutants in tomato plants, highlighting the need for careful selection and validation of target genes for genome editing. CRISPR-Cas9′s versatility extends to metabolic engineering, where it is used to modify plant cells to produce specific metabolites and improve various agronomic traits, including stress tolerance [95,96]. Its application in creating drought-resistant varieties in crops like weedy rice [97] and improving plant architecture and drought tolerance in maize [98] exemplifies its potential to enhance crop resilience and productivity. The efficacy of CRISPR/Cas9 in inducing targeted genetic modifications has been underscored by LeBlanc et al. [99], who demonstrated its enhanced efficiency under heat stress conditions, subsequently improving the drought stress response in plants. This finding is further corroborated by Chen et al. [93], who observed that the knockout of AITR genes in *A. thaliana* via CRISPR/Cas9 not only conferred enhanced drought tolerance but also did so without imposing fitness costs, indicating the technology’s potential for precise stress tolerance enhancement without adversely affecting plant growth or productivity. Kumar et al. [100] extended these observations to *Solanum lycopersicum* (tomato), where CRISPR/Cas9-edited mutants exhibited improved drought stress responses, further validating the utility of this technology in enhancing drought tolerance across different plant species. Tiwari et al. [101] expanded the scope of CRISPR/Cas9 applications by editing GRXS14/15/16/17 genes in tomato, achieving increased tolerance not only to drought but also to a spectrum of abiotic stresses, thereby highlighting the technology’s capacity for instilling multi-stress resilience in crops. The integrative approach combining omics and gene editing, as emphasized by Razzaq et al. [102], delineates a comprehensive strategy for developing stress-tolerant crops, signifying the broader impact of CRISPR/Cas9 in addressing various plant stressors. The specific involvement of OsmiR818b in drought response, as elucidated through CRISPR/Cas9-mediated mutagenesis by Chung et al. [103] in rice, exemplifies the technology’s precision in dissecting the genetic underpinnings of stress responses. Krishna et al. [104] further highlight CRISPR/Cas9′s role in conferring molecular immunity against drought stress in tomato, underlining its effectiveness in bolstering plant resilience. The versatility of CRISPR/Cas9 is also evident in the research of Zheng et al. [105], who demonstrated its application in enhancing stress resistance across different plant species, thereby showcasing its broad applicability in plant stress management. Moreover, Park et al. [106] presented evidence of CRISPR/Cas9′s capability to improve both abiotic and biotic stress tolerance in rice, underscoring its potential as a comprehensive tool for enhancing plant resilience to a wide array of environmental challenges, including drought. Collectively, these studies affirm the transformative potential of CRISPR/Cas9 technology in plant science, particularly in engineering drought tolerance. The specificity, efficiency, and versatility of CRISPR/Cas9 not only facilitate a deeper understanding of the genetic mechanisms underlying stress responses but also pave the way for the development of crops with enhanced resilience to environmental stressors, thereby contributing to sustainable agriculture and food security.

Moreover, the technology has been applied to engineer resistance against a range of plant diseases, including geminiviruses in cassava [107] and tomato yellow leaf curl virus (TYLCV) in tomato plants [108], showcasing its potential in conferring durable disease resistance in crops. In summary, the CRISPR-Cas9 system represents a powerful tool in plant biotechnology, offering promising applications in developing drought-resistant plant varieties, enhancing stress tolerance, engineering disease resistance, and improving plant architecture. As research in this field continues to evolve, CRISPR-Cas9 is poised to play a pivotal role in addressing some of the most pressing challenges in agriculture, contributing to developing crops more resilient to environmental stresses and diseases. However, careful selection and understanding of target genes are imperative to harness this technology’s potential in crop improvement fully.

## 6. Role of Microbial Biotechnology in Enhancing Drought Tolerance

Plant growth-promoting rhizobacteria (PGPR) and mycorrhizae, the critical components of the soil microbiome, have garnered significant attention for their role in enhancing plant growth and health. These beneficial soil microorganisms offer a natural and sustainable alternative to chemical fertilizers in agriculture, underlining their importance in modern agronomic practices. Mycorrhizae, particularly arbuscular mycorrhizal fungi (AMF), are renowned for promoting plant growth and aiding in its resilience against biotic and abiotic stresses. This is achieved mainly through enhancing nutrient uptake, especially phosphorus, which is crucial for plant growth [109,110]. Mycorrhizal infection has been shown to positively impact the mineral nutrition of plants, facilitating growth through the fungus’s efficient nutrient acquisition [111].

Additionally, mycorrhizae can induce changes in plant defensive chemistry, offering biological resistance against pathogens and herbivores, albeit to a lesser extent than direct responses to these threats [112]. Similarly, PGPR is instrumental in improving plant mineral nutrition, thus suggesting their potential application as biofertilizers [113]. The successful use of PGPR in agriculture is closely linked with the reciprocal gene regulation between these bacteria and plants during colonization, indicating a synergistic interaction that benefits plant growth [114]. PGPR is known for both direct and indirect positive effects on plant health, which include enhancing nutrient availability and providing phytohormones and other growth stimulants [115,116]. The introduction of PGPR and mycorrhiza into nutrient-poor soils has increased microbial activity and soil element availability, thereby improving soil quality and fostering plant growth [117].

Additionally, the use of these microorganisms has been associated with alterations in the plant’s primary and secondary metabolome, contributing to enhanced growth and yield [118]. Significantly, PGPR suppresses plant diseases and directly improves plant health by enhancing nutrient availability and acting as a phytostimulant [119]. The interaction between PGPR and plants is complex and can be influenced by factors such as crop genotype and the presence of symbiotic fungi. This complexity underscores the intricate nature of their interactions with plants [120]. Furthermore, PGPR has been suggested to restore soil fertility, leveraging the nitrogen-fixing abilities of certain bacteria within this group [121].

On the other hand, mycorrhizae play a vital role in increasing nutrient absorption, synthesizing plant growth hormones, and enhancing plant resistance to drought stress, thereby improving plant resilience to environmental challenges [122]. Mycorrhizae and PGPR have been proposed as environmentally friendly alternatives to traditional fertilization methods, offering the potential for reduced environmental impact and biofuel production [123]. In summary, both PGPR and mycorrhizae are invaluable in sustainable agriculture, offering a range of benefits from enhancing plant growth and nutrient uptake to bolstering resilience against biotic and abiotic stresses. Their role as biofertilizers and soil quality improvers positions them as critical tools in pursuing sustainable and environmentally friendly agricultural practices.

The diverse roles of plant growth-promoting rhizobacteria (PGPR) and mycorrhizae in enhancing drought tolerance in various plant species have been elucidated through multiple studies. As described in Table 3, these studies cover a spectrum of plant species and experimental approaches. Collectively, they highlight how PGPR and mycorrhizae contribute to improved drought resilience through mechanisms like altering antioxidant activities, enhancing nutrient and water uptake, and modifying physiological and biochemical responses. Table 4 serves as a comprehensive summary of the current understanding in this field, offering valuable insights into the potential application of these microorganisms in mitigating drought stress in agriculture.

## 7. Omics Approaches in Drought Stress Research

A comprehensive understanding of plant responses to drought stress is crucial for developing effective strategies to enhance crop drought tolerance. Integrating genomics, proteomics, and metabolomics has significantly advanced our understanding of the complex molecular processes involved in drought responses. Proteomic analyses have been instrumental in revealing significant changes in various pathways under drought stress, including sensing and signal transduction, reactive oxygen species scavenging, osmotic regulation, gene expression, protein synthesis/turnover, and carbohydrate and energy metabolism [135]. These studies underscore the complex nature of drought tolerance, which involves many signaling mechanisms and molecular responses that are differentially expressed under stress conditions [136]. The simultaneous analysis of metabolism and nutrient stoichiometries in plant shoots and roots has provided insights into contrasting responses to experimental drought and seasonally changing conditions [137]. Metabolomic technologies have further emphasized metabolites’ role in drought responses and plant tolerance, offering a detailed understanding of the biochemical changes that occur during drought stress [138]. High-throughput phenotyping methods, including quantitative trait locus (QTL) mapping and genome-wide association studies (GWAS) using next-generation sequencing (NGS) technologies, have enabled a more in-depth study of physiological responses and underlying molecular mechanisms in crops under drought stress [139]. Research into the genomic background and molecular mechanisms of drought responses in wild wheat progenitors has revealed favorable regulatory elements that may have been lost during domestication and cultivation processes [140].

The significance of proteomics in understanding the complex molecular mechanisms of drought stress response in crucial crops like rice, maize, and wheat has been highlighted, demonstrating how proteins drive cellular events and adaptative processes during drought stress [141,142]. Transcriptomics and metabolomics integration have provided valuable insights into the drought tolerance mechanisms in quinoa, highlighting the molecular responses of drought-tolerant genotypes [143].

Genetic control of drought tolerance in soybeans has been identified, emphasizing the importance of QTLs and specific genes associated with drought stress responses [144]. Genomic variation in *Brachypodium* reflected as distinctive metabolomes indicates the potential mediation of drought tolerance through metabolomic variations [145]. Further research into different genotypes of Populus has shed light on intra-specific variation in drought responses, with studies comparing proteome responses under dehydration shock and cyclic post-drought re-watering, thereby providing comprehensive insights into the proteomic adaptations in response to drought stress [146,147].

Metabolomic approaches have also been employed to analyze the molecular differences between drought-sensitive and drought-tolerant poplar species, demonstrating metabolomics’ potential in unraveling drought resistance mechanisms [148]. Studies on other plants like Morus alba L. and cucumber have also contributed to our understanding of plant responses to short-term drought and re-watering through morphological structure and proteomic analyses [149,150].

In conclusion, integrating genomics, proteomics, and metabolomics has significantly enriched our understanding of plant responses to drought stress. These multidisciplinary approaches have opened new avenues for identifying molecular mechanisms underlying drought tolerance, offering potential targets for developing drought-resistant crops and contributing to sustainable agricultural practices.

## 8. Challenges and Future Perspectives

The current landscape of biotechnological interventions for enhancing crop drought tolerance presents a complex and multifaceted challenge. Advancements in understanding plant responses to drought stress at both molecular and whole-crop levels have been significant, yet the task remains intricate due to the concurrent occurrence of various abiotic stresses like high temperatures, high irradiance, and nutrient deficiencies [61,151]. Developing drought-resistant varieties through traditional breeding complemented by biotechnological tools such as identifying drought-resistant genes, QTL analysis, gene transformation, and marker-assisted selection is crucial in this endeavor [151]. The role of biotechnology in enhancing plant tolerance to drought stress is increasingly relevant, as is the need to develop crops that can withstand challenging environmental conditions [152,153]. The engineering of plants for improved tolerance to abiotic stresses, including drought, is a significant objective of plant biotechnology and is anticipated to have commercial applications shortly [28]. However, engineering these traits is challenging due to the complexity of native stress responses in plants [154].

Recent research has focused on uncovering new strategies and potential breakthroughs of stress management in plant drought. The role of microRNAs (miRNAs) in enhancing plant drought tolerance offers a promising avenue for future research, particularly in improving cereal crops for drought tolerance [155]. Additionally, the application of silicon has shown potential in improving water status, photosynthesis, and mineral nutrient absorption in rice plants under drought stress, suggesting a new strategy to alleviate drought stress in plants [156].

Emerging research is also linking molecular responses in drought-stressed plants to broader ecological contexts, such as tripartite species interactions and the ecology of insect-transmitted pathogens, which is particularly relevant in the era of climate change [157]. Understanding the influence of plant hormones on drought stress responses and phytohormonal interactions is another critical area, with the potential to manipulate phytohormones to enhance plant drought tolerance [158]. Furthermore, increasing the research on plant responses to combined high temperature and drought stress is essential in future climate change, emphasizing the need to regulate the growth and development of food crops under these stress conditions [159].

Innovative techniques like acoustic emissions for measuring drought-induced cavitation in plants are opening new research frontiers, potentially increasing our understanding of drought tolerance and recovery capabilities [160]. Additionally, studies on the impact of drought stress on plant susceptibility to herbivore colonization, such as in *Solanum lycopersicum*, shed light on the complex interactions between drought stress and plant–herbivore relationships, an important consideration in understanding the ecological implications of drought stress [161].

In conclusion, synthesizing these diverse studies underscores the potential breakthroughs and future research directions in plant drought stress. These include the exploration of miRNAs, silicon application, plant–vector–pathogen interactions, phytohormonal responses, the challenges of combined high temperature and drought stress, novel techniques for studying drought tolerance, and the ecological implications of drought stress on plant–insect interactions. This multifaceted approach, blending traditional and biotechnological strategies, is essential for advancing our understanding and managing drought stress in crops.

## 9. Conclusions

This article concludes that enhancing drought tolerance in crops through biotechnological interventions is a multifaceted challenge that requires a comprehensive understanding of plant responses to drought stress. Integrating genomics, proteomics, metabolomics, and advanced genetic engineering tools like CRISPR-Cas9 has significantly enriched our knowledge in this field. Moreover, the role of microbial biotechnology in improving plant resilience against drought stress further emphasizes the potential of these interventions. Combining traditional breeding methods and modern biotechnological approaches offers promising prospects for developing drought-tolerant crops, contributing to sustainable agricultural practices, and ensuring food security under changing climatic conditions.

## Figures and Tables

**Table 1 plants-13-00717-t001:** Genes that have been identified as crucial in drought response.

Gene	Plant Species	Function	Observed Effects	References
ARF, DREB, WRKY, NAC, TCP, MYB, GRAS family transcription factors, abscisic acid, dehydrin-related genes	*Dactylis glomerata* L.	Post-transcriptional gene regulation	Targeting a wide range of drought-related genes	[24]
Transcription factors, protein phosphatase 2Cs, late embryogenesis abundant proteins	*Glycine max* L.	Tolerance responses to abiotic stresses	Elevated levels of transcripts under recurrent drought conditions	[25]
TLP5, NSP2, DRE1D, NAC29, TLP 3, HFA6B	*Gossypium hirsutum*	Abiotic stress-responsive elements	Abundance of MYB binding sites involved in drought-inducibility	[26]
MaWRKYIII8	*Morus*	Response to drought and abscisic acid	Important role in mulberry response to drought stress	[27]
ABF3	*Arabidopsis thaliana*	Reprogramming of drought response	Changes in the timing or strength of expression of some drought response genes	[28]
GRMZM2G546097, contig854, contig4549, contig4777, contig312, contig6971, contig7875	Tropical Maize	Delay of flowering time under drought stress	Responsive to drought and expected to play roles analogous to those of drought-responsive genes	[29]
ROS-scavenging gene families (ascorbate peroxidase, monodehydroascorbate reductase, peroxiredoxin)	*Medicago sativa*	Scavenging reactive oxygen species	Conserved, tissue-specific patterns of gene expression in response to drought	[30]
CIPK Families	Rice, Maize, Sorghum	Drought responsiveness	Confirmed drought responsiveness and conservation of functions between species	[31]
At1g08710	*Arabidopsis thaliana*	Negative regulation of root length and drought stress tolerance	Activation of drought stress-responsive genes such as RD29A/B, COR47, KIN1	[32]

**Table 2 plants-13-00717-t002:** Physiological and molecular responses of plants to drought stress.

Response Category	Specific Changes/Responses	Description/Details
Stomatal Closure	Reduction in transpiration	Regulated by abscisic acid, conserves water by reducing water loss through leaves.
Root System Alteration	Enhanced root growth	Increased root length and density, changes in morphology to improve water uptake from deeper soil layers.
Osmotic Adjustment	Accumulation of osmolytes	Compounds like proline, glycine betaine, and sugars lower osmotic potential, aiding in water retention and absorption.
Leaf Morphology Changes	Adaptations in leaf structure	Modifications such as reduced leaf size, thicker leaves, and a waxy surface to minimize transpiration and enhance water retention.
Photosynthesis Reduction	Decrease in photosynthetic activity	Limited CO_2_ availability due to stomatal closure and potential damage to photosynthetic apparatus under stress.
Molecular Responses	Gene regulation	Upregulation or downregulation of drought-responsive genes controlled by transcription factors like DREB, NAC, and MYB.
Protein Synthesis and Turnover	Changes in protein profiles	Increased synthesis of protective proteins like heat shock proteins and LEA proteins.
ROS Scavenging	Enhanced antioxidant defense	Activation of enzymes like superoxide dismutase, catalase, and peroxidase to mitigate oxidative stress caused by reactive oxygen species.
Secondary Metabolites Accumulation	Increase in protective compounds	Accumulation of flavonoids and anthocyanins, contributing to protection against oxidative stress and UV radiation absorption.

**Table 3 plants-13-00717-t003:** Comparison of traditional breeding methods and biotechnological approaches in crop improvement.

Aspect	Traditional Breeding Methods	Biotechnological Approaches
Methodology	Involves cross-breeding and selection based on phenotypic traits.	Utilizes molecular tools like gene editing, QTL mapping, and marker-assisted selection.
Timeframe	Time-consuming; often takes several years to develop new varieties.	Relatively faster; can produce results in a shorter period.
Precision	Less precise; dependent on natural genetic variations.	High precision in targeting specific genes or traits.
Outcome Diversity	Limited by genetic variability within accessible germplasm.	It can introduce novel traits not present in the gene pool, including those from different species.
Resource Intensity	Generally less resource-intensive but dependent on trial and error.	More resource-intensive, requiring advanced technological inputs and expertise.
Adaptability to Stress	Limited by existing genetic variation in response to stresses.	Enhanced ability to develop stress-resistant varieties, including drought and disease resistance.
Commercialization	Slower due to lengthy breeding cycles.	Faster potential for commercialization due to rapid development.
Ecological Impact	Relatively lower ecological impact.	Potential risks and ethical concerns due to genetic modifications.

**Table 4 plants-13-00717-t004:** Impact of PGPR and mycorrhizae on plant drought tolerance: a review of recent studies.

Reference	Species Studied	Methods	Key Findings
Morcillo and Manzanera [124]	Plants in General	Review of Literature	Applying PGPR enhances drought tolerance by altering antioxidant activity under water-deficit conditions.
Khan et al. [125]	Chickpea (*Cicer arietinum* L.)	Physiological and Biochemical Analysis	Explored the role of PGPR and PGRs on chickpea physiology and biochemistry under drought stress, associating it with improved drought tolerance.
Naseem and Bano [126]	Maize	Isolation and Characterization of EPS Produced by PGPR	Examined the drought tolerance potential of PGPR on maize, used alone or in combination with their respective EPS.
Cosme [127]	Plants in General	Evolution of Plant Adaptation to Drought	Suggested that mycorrhizas might accelerate evolutionary gains in drought tolerance due to the success of mycorrhizal plants in repeated drought conditions.
Zhao et al. [128]	Plants in General	Meta-Analysis	Found that members of the order Bacillales (PGPR) enhance drought tolerance through their spore-forming ability.
Subramanian et al. [129]	Plants in General	Literature Review	Mycorrhiza reportedly supports plant growth under inhospitable weather conditions, with variability in drought tolerance intensity and genotype potential.
Spinoso-Castillo et al. [130]	Sugarcane (*Saccharum* spp.)	Ex Vitro Acclimatization Experiment	Demonstrated that symbiotic associations between arbuscular mycorrhizal fungi and plants induce drought stress tolerance.
Vílchez et al. [131]	Plants in General	Trehalose Production by Desiccation-Tolerant Microorganisms	Indicated that PGPR presence in roots increases crop production by enhancing plant tolerance to drought and other stresses.
Qiao et al. [132]	Pigeon Pea	Investigation of AM Effects on Drought Tolerance	Observed that arbuscular mycorrhizae enhance the drought tolerance of pigeon pea and trigger physiological responses to water deficit.
Farnia and Khodabandehloo [133]	Maize (*Zea mays* L.)	Foliar Application of Zinc Nutrient and Mycorrhiza Under Water Stress	Found that mycorrhiza increases nutrient uptake, salinity tolerance, drought tolerance, water uptake, root disease resistance, and photosynthesis in maize.
Gusain et al. [134]	Rice (*Oryza sativa* L.)	Bacteria-Mediated Amelioration of Drought Stress	Demonstrated that PGPR application enhances the drought tolerance of rice under water-deficit conditions.

## Data Availability

Not applicable.

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
