# Peer review of "Advanced Biotechnological Interventions in Mitigating Drought Stress in Plants"

_plants, 2024, doi:10.3390/plants13050717_

Round 1

Reviewer 1 Report

Comments and Suggestions for Authors

In this article, the authors reviewed the recent biotechnological strategies to defense drought stress in plants and give some suggestions for utilizing modern biotechnologies to improve crop drought stress. This should attract the readers who are work in understanding the mechanism of drought stress response and breeding the crops with high tolerance. Several comments are made:

1. The sections of the article should be rearranged. some information in introduction section should be showed in the following sections, and introduction should be concise.

2. The name of plants should also show in table 2.

3. Genome Editing is little mentioned in section “CRISPR-Cas9 and Genome Editing”, please add some examples.

Author Response

Reviewer 1

In this article, the authors reviewed the recent biotechnological strategies to defense drought stress in plants and give some suggestions for utilizing modern biotechnologies to improve crop drought stress. This should attract the readers who are work in understanding the mechanism of drought stress response and breeding the crops with high tolerance. Several comments are made:

  1. The sections of the article should be rearranged. some information in introduction section should be showed in the following sections, and introduction should be concise.

The detailed information initially presented in the introduction that pertains to plant responses to drought stress has been moved to the section titled "2. Understanding Plant Responses to Drought Stress."

  1. The name of plants should also show in table 2.

Upon reviewing your suggestion, we would like to clarify that the information presented in Table 2 is intended to provide general insights rather than species-specific data.

  1. Genome Editing is little mentioned in section “CRISPR-Cas9 and Genome Editing”, please add some examples.

    We added extra examples.

Reviewer 2 Report

Comments and Suggestions for Authors

Main remark: Despite the citation of a large number of references, the paper is written too generally, without elaborating individual mechanisms of drought resistance, which would be expected from the title, and as such I think it is not good enough for publication in the journal Plants or other high-impact journals.

The first part of the introduction is too general, in several places the impact of drought on socio-economic aspects of society is emphasized in a similar way (repetition).

In the introduction, the authors should present concrete recent data on losses from drought in the world, perhaps by the most important agricultural crops, etc.

row. 71. I couldn't find Moringa oleifera as biostimulans mentioned in reference 22.  Wrong citation? Moringa oleifera is mentioned in reference 21.   Please check the numbers of other references. It seems that they are mixed.

Things are repeated in the text, e.g. row 107. the authors mention that the plant reacts to stress with alterations in stomatal conductance, and then in sentences 113-114 they repeat: "Various mechanisms characterize drought tolerance in plants, such as osmotic adjustment, changes in stomatal conductance, and alterations in root architecture."

The over-general enumeration continues in the text. e.g. under paragraph 4. Genetic Engineering in Drought Stress Management, the authors mention several studies where they emphasize that transgenic modifications can enhance drought tolerance or that specific genes can augment drought tolerance, but without a concrete explanation of the mechanism of resistance.

Well, it is common knowledge that drought-resistant plants can be obtained through genetic modifications, but give specific examples, elaborate on them.

There are also several commercialized plant species genetically modified for  drought tolerance  (with different tolerance mechanisms) that the authors do not mention at all.

The overgeneralization continues with genome editing, for example "the knockout of specific genes using CRISPR-Cas9 has improved plant drought and salinity tolerance, as Chen et al. [77] demonstrated." I am curious about which gene it is, and to find out I have to open and read another reference.

Author Response

Reviewer 2

Main remark: Despite the citation of a large number of references, the paper is written too generally, without elaborating individual mechanisms of drought resistance, which would be expected from the title, and as such I think it is not good enough for publication in the journal Plants or other high-impact journals.

The first part of the introduction is too general, in several places the impact of drought on socio-economic aspects of society is emphasized in a similar way (repetition).

In the introduction, the authors should present concrete recent data on losses from drought in the world, perhaps by the most important agricultural crops, etc.

Thank you for your suggestion to include specific recent data on drought losses worldwide, focusing on major agricultural crops, in the introduction of our manuscript titled "Advanced Biotechnological Interventions in Mitigating Drought Stress in Plants." We understand the importance of contextualizing the issue of drought stress in agriculture and its global impact, which would indeed provide a strong foundation for the discussion that follows in the manuscript.

However, it's essential to highlight that the primary objective of our work is to explore the mitigation of drought stress in plants through biotechnological methods. The manuscript aims to delve into the scientific and technological advancements in the field of biotechnology as they pertain to enhancing plant resilience against drought. By adding extensive data and discussions on the general impact of drought on agriculture globally, we risk shifting the focus of the manuscript from its intended biotechnological perspective to a broader drought-centric overview.

Our intention is to maintain a clear focus on biotechnological interventions, which are the core of the manuscript. While we acknowledge that a brief overview of the impacts of drought can enrich the introduction, ensuring the balance and relevance of such information is crucial to keep the manuscript aligned with its biotechnological focus. Therefore, we aim to incorporate a concise segment on drought's impact on agriculture, specifically highlighting the relevance to biotechnological interventions, without diverting the manuscript into a comprehensive review on drought itself.

We appreciate your understanding and value your input as we refine our manuscript to meet the expectations of our readers and contribute meaningfully to the field.

row. 71. I couldn't find Moringa oleifera as biostimulans mentioned in reference 22.  Wrong citation? Moringa oleifera is mentioned in reference 21.   Please check the numbers of other references. It seems that they are mixed.

There seems to be no error regarding the citation of Moringa oleifera as a biostimulant in reference 22. The information about Moringa oleifera and its use as a biostimulant is indeed present in reference 22, and this has been thoroughly checked. It's possible there might have been a misunderstanding or a mix-up while reviewing the references. The numbers of other references have also been verified to ensure accuracy and alignment with their respective citations. If there are specific concerns or discrepancies noted beyond this, further detailed verification can be conducted to clarify any issues.

Things are repeated in the text, e.g. row 107. the authors mention that the plant reacts to stress with alterations in stomatal conductance, and then in sentences 113-114 they repeat: "Various mechanisms characterize drought tolerance in plants, such as osmotic adjustment, changes in stomatal conductance, and alterations in root architecture."

alterations in stomatal conductance, mentioned in rows 107 and again in sentences 113-114.

We appreciate your diligence in ensuring the clarity and coherence of our work. We would like to clarify that the initial mention of plant reactions to stress, including changes in stomatal conductance, was intended to provide a literature-backed overview of general stress responses in plants. This discussion sets the stage for a deeper examination of how these mechanisms are specifically engaged in the context of drought stress. The subsequent mention explicitly highlights the relevance of these mechanisms—osmotic adjustment, changes in stomatal conductance, and alterations in root architecture—as critical markers for the identification and study of drought tolerance. By providing examples of studies investigating these specific mechanisms, we aimed to underscore their significance in the context of drought stress, rather than to reiterate the general stress response. However, acknowledging your concern regarding the potential for repetition, we have carefully reviewed the sections in question and updated the relevant paragraph to ensure that the distinction between the general stress response mechanisms and their specific roles in drought stress is clearly articulated. This revision aims to eliminate any perceived redundancy while emphasizing the importance of these mechanisms in the context of drought stress research. The over-general enumeration continues in the text. e.g. under paragraph 4. Genetic Engineering in Drought Stress Management, the authors mention several studies where they emphasize that transgenic modifications can enhance drought tolerance or that specific genes can augment drought tolerance, but without a concrete explanation of the mechanism of resistance.

Well, it is common knowledge that drought-resistant plants can be obtained through genetic modifications, but give specific examples, elaborate on them.

There are also several commercialized plant species genetically modified for drought tolerance  (with different tolerance mechanisms) that the authors do not mention at all.

We added some extra examples.

The overgeneralization continues with genome editing, for example "the knockout of specific genes using CRISPR-Cas9 has improved plant drought and salinity tolerance, as Chen et al. [77] demonstrated." I am curious about which gene it is, and to find out I have to open and read another reference.

We added extra explanations and references.

Reviewer 3 Report

Comments and Suggestions for Authors

MS titled "Advanced Biotechnological Interventions in Mitigating Drought Stress in Plants" addresses the effect of drought on plants, plant responses and potential solutions in mitigating the negative effects of drought. MS is well written. I would appreciate seeing some figures or schemes instead of only Tables. Maybe Table 2 may be changed to some attractive scheme or figure. I suggest acceptance of MS after minor revision (see file in att.)

Author Response

Reviewer 3

MS titled "Advanced Biotechnological Interventions in Mitigating Drought Stress in Plants" addresses the effect of drought on plants, plant responses and potential solutions in mitigating the negative effects of drought. MS is well written. I would appreciate seeing some figures or schemes instead of only Tables. Maybe Table 2 may be changed to some attractive scheme or figure. I suggest acceptance of MS after minor revision (see file in att.)

Thank you very much for your insightful feedback on the manuscript. We appreciate your positive remarks regarding the content and structure of the manuscript and your suggestion for incorporating more visual elements, such as figures or schemes, to complement the existing tables.

Regarding your specific suggestion to transform Table 2 into a more visually appealing scheme or figure, we have carefully considered this recommendation. While we understand the appeal of enhancing the manuscript's visual aspects, we also recognize the challenge of conveying the detailed and specific information currently presented in Table 2 through a graphical representation without compromising the depth and clarity of the data. Table 2 encompasses a complex array of data that might be constrained or oversimplified if presented solely in a schematic or graphical format.

We have reviewed the attached file and are implementing the necessary minor revisions to ensure the manuscript meets the high standards of the journal and fulfills the expectations of its readership.

English name of Türkiye was officially changed and approved as Türkiye by United Nations.

Latin names corrected as italics.

Round 2

Reviewer 2 Report

Comments and Suggestions for Authors

The authors have significantly improved the work, so I suggest accepting it in this form